# Automated Assessment of Comprehension Strategies from Self-Explanations Using LLMs

Bogdan Nicula [1], Mihai Dascalu [1,2,*] , Tracy Arner [3] , Renu Balyan [4] and Danielle S. McNamara [3]

1 Computer Science and Engineering Department, National University of Science and Technology POLITEHNICA of Bucharest, 313 Splaiul Independentei, 060042 Bucharest, Romania; bogdan.nicula@upb.ro
2 Academy of Romanian Scientists, Str. Ilfov, Nr. 3, 050044 Bucharest, Romania
3 Department of Psychology, Arizona State University, P.O. Box 871104, Tempe, AZ 85287, USA; tarner@asu.edu (T.A.); dsmcnama@asu.edu (D.S.M.)
4 Math/CIS Department, SUNY Old Westbury, Old Westbury, NY 11568, USA; balyanr@oldwestbury.edu
* Correspondence: mihai.dascalu@upb.ro

**Abstract:** Text comprehension is an essential skill in today's information-rich world, and self-explanation practice helps students improve their understanding of complex texts. This study was centered on leveraging open-source Large Language Models (LLMs), specifically FLAN-T5, to automatically assess the comprehension strategies employed by readers while understanding Science, Technology, Engineering, and Mathematics (STEM) texts. The experiments relied on a corpus of three datasets (N = 11,833) with self-explanations annotated on 4 dimensions: 3 comprehension strategies (i.e., bridging, elaboration, and paraphrasing) and overall quality. Besides FLAN-T5, we also considered GPT3.5-turbo to establish a stronger baseline. Our experiments indicated that the performance improved with fine-tuning, having a larger LLM model, and providing examples via the prompt. Our best model considered a pretrained FLAN-T5 XXL model and obtained a weighted F1-score of 0.721, surpassing the 0.699 F1-score previously obtained using smaller models (i.e., RoBERTa).

**Keywords:** language models; large language models; self-explanation; self-explanation strategies

## 1. Introduction

Reading and learning from text are critical skills for learners to acquire new knowledge, which is essential for educational and career success. To comprehend text, the reader constructs a mental model of the text while he/she reads. This mental model can be represented at three levels: (1) surface-level knowledge of the exact words in the text, (2) the textbase-level semantic representation of ideas, and (3) the situation model, which combines the textbase with the reader's prior knowledge. The ability to leverage strategies that support comprehension is a critical skill that readers need in the absence of the essential prior knowledge necessary to develop a coherent situation model. Proficient readers are more likely to spontaneously employ strategies while reading to help them comprehend difficult texts than students who are less-skilled readers [1]. Fortunately, students can learn when and how to implement these reading comprehension strategies through direct instruction and deliberate practice. One such strategy, with considerable evidence supporting its use by students with limited prior knowledge or lower reading skills, is self-explanation.

Self-Explanation (SE) is the practice of explaining the meaning of portions of a text to one's self while reading. Engaging in self-explanation encourages students to generate inferences, in which they connect sentences or idea units between text sections or texts. Similarly, students may generate elaborative self-explanations in which they connect their prior knowledge to new information they read in the text. Generating bridging and

elaborative self-explanations supports readers' inference making, which, in turn, supports the development of their mental representation of the text.

Developed by McNamara [2], Self-Explanation Reading Training (SERT) teaches readers strategies to enhance text comprehension. The training guides students through each strategy in increasing order of difficulty, starting with comprehension monitoring. Comprehension monitoring aims to help students understand when they need to implement the remaining strategies to support their comprehension. This work focused on the three remaining strategies: paraphrasing, bridging inference, and elaboration.

The paraphrasing strategy refers to reformulating a sequence of text in one's own words. SERT can help develop readers' text comprehension skills by prompting them to access their vocabulary to translate the ideas into more-familiar language. Bridging involves linking multiple ideas across a text or across multiple texts (e.g., two different articles about the same topic). Generating bridging inferences requires the reader to find connections between the ideas and to structure them in a coherent way. Elaboration involves linking information in the text and the reader's knowledge base; this helps the reader integrate new information with existing knowledge. Collectively, these strategies support readers' construction of more-coherent mental representations of the text, in particular challenging texts that require substantial prior knowledge to understand.

Considerable evidence indicates that these strategies support readers' comprehension of complex texts. However, additional benefits can be realized when the reader receives feedback about the accuracy or quality of his/her self-explanation [3]. One way readers can receive feedback is from instructors who review and score self-explanations based on a rubric [4]. This method is time-consuming and does not provide readers with the feedback they need in real-time. To alleviate this challenge, students can practice their reading and self-explaining using an intelligent tutoring system, where they both have the opportunity to engage in the deliberate practice of reading and self-explaining, but they also receive essential guiding feedback [5]. Thus, refining and improving software applications that can detect the presence of these strategies in the readers' constructed responses can be helpful for both evaluation and training. Natural Language Processing (NLP) [6] techniques and Machine Learning can be used to develop such models, given a large enough dataset containing labeled examples of the presence and absence of these strategies in readers' self-explanations. Previous work [7] has shown that such automated models can be built to reliably assess self-explanation reading strategies. The recent release of more-sophisticated and readily accessible large language models further supports the expansion of this prior work.

## 1.1. Large Language Models

Large Language Models (LLMs) have recently gained notoriety through their association with popular chatbot systems such as OpenAI's ChatGPT [8] or Google's Bard [9]. These models are trained on massive amounts of heterogeneous text data (including news articles, web pages, social media posts, and scanned books) and datasets tailored to specific tasks. They manage to capture statistical patterns of natural language, such as syntax, semantics, and pragmatics. Their knowledge of these patterns enables the generation of new complex texts relevant to the input with which they have been prompted. LLMs are highly adaptable to different NLP tasks and domains and can be fine-tuned on specific datasets or prompts to perform a wide variety of natural-language-generation tasks, including summarizing, translation, text completion, and question answering. They also manifest "emergent capabilities" [10], skills they were not trained explicitly on, but are easy to solve based on the memorized statistical patterns.

LLMs are a fairly recent type of neural architecture that has grown in size and performance in the past few years. They are part of a class of deep learning architectures called Transformers, stemming from the original model introduced by Google in 2017 [11]. Depending on their structure, modern Transformer-based models can be classified into three categories:

- **Encoder only:** models that understand the text and are used in classification/ regression tasks. An example of an encoder-only model is the Bidirectional Encoder Representations from Transformers model, or BERT [12], followed by its improved version, Robustly Optimized BERT Pretraining Approach (RoBERTa) [13].
- **Decoder only:** models that excel at text generation. The Generative Pretrained Transformer (GPT) family with various versions (e.g., 3 [14] or 4 [15]) are good examples of the decoder-only architecture.
- **Encoder–decoder:** models capable of both understanding and generating text. They are useful for translation, abstractive summarization, question answering, and many other tasks. The Text-to-Text Transformer (T5) [16], followed by its improved version, Fine-tuned Language Net (FLAN-T5) [17], pretrained on a large collection of datasets, are examples of such architectures.

However, LLMs' impressive capability to generate various relevant, cohesive, and coherent texts comes with caveats. These models can sample from the most-statistically relevant sequences and complete a given prompt flawlessly. Still, they do not offer guarantees regarding the correctness of the generated information [18]. Furthermore, they are still susceptible to a variety of attacks, such as injecting a request with a small sequence of words that can deviate the flow of the interaction in a different direction from what was intended initially [19].

### 1.1.1. FLAN T5

The T5 model is an encoder–decoder Transformer trained on a combination of supervised and unsupervised tasks, all having a text-to-text format (i.e., receiving text input and outputting text). The supervised training is performed on tasks from the General Language Understanding Evaluation (GLUE) [20] and SuperGLUE [21] benchmarks converted to fit the text-to-text paradigm. The unsupervised or self-supervised tasks involve reconstructing the original text when receiving corrupted input (e.g., by randomly removing 15% of tokens and replacing them with sentinel tokens). The T5 models that have been made public cover a wide range of sizes, from the 60-million-parameter T5-small model to the 11-billion-parameter T5-11b model.

The FLAN-T5 model [17] represents an enhanced version of T5 fine-tuned on a larger number of tasks while emphasizing chain-of-thought scenarios. Using the FLAN approach, the authors trained both T5 and a Pathways Language Model (PaLM) [22] and achieved state-of-the-art performance on several benchmarks with the 540-billion-parameter FLAN-PaLM model.

### 1.1.2. GPT

Generative Pretrained Transformer(GPT) models are a family of decoder Transformer-based models [23]. They consist only of decoder blocks and are left-to-right autoregressive models. The first model, GPT-1, consisted of a 117-million-parameter network pretrained in an unsupervised setting and, then, fine-tuned on individual tasks. The pretraining procedure was a classical language modeling task in which the model had to predict the likeliest sequence of words, given a fixed input sequence. After pretraining, the model could be used for various tasks, including classification, paraphrase identification, or question answering.

The GPT-2 model was a 1.5-billion-parameter model [24]. It relied on a similar pretraining approach where the likeliest sequence given the current input and task must be predicted by the model. At the time of its release, the model obtained state-of-the-art performance in 7 out of the 8 tested language-modeling tasks, without task-related fine-tuning.

The GPT-3 model was perceived as a considerable step forward when released in 2020 in terms of performance and size (175-billion parameters). The GPT3.5-turbo model was released in November 2022, with OpenAI providing scarce information regarding its training. Its size is estimated to be comparable to that of GPT3.5. The model was trained

using Reinforcement Learning from Human Feedback (RLHF) [25] and is designed to perform better in conversational settings and iterative task-solving. It gained popularity as it represented the backbone of the popular free version of the ChatGPT conversational agent.

Besides GPT3.5-turbo, OpenAI provides several other text-to-text endpoints such as "text-davinci-003" and GPT4 [15]. The former was tested on a subset of the tasks presented in this article, but did not perform better than the GPT3.5-turbo endpoint. The latter, GPT4, is the backbone of the paid ChatGPT Plus service and is expected to provide better replies. Still, it is also a more-closed system with little detail being provided about its architecture, the training dataset, and/or the training setup. We opted against evaluating this alternative as, when the experiments were performed, it was 20–30× more expensive than GPT3.5-turbo. Although GPT4 obtains better performances and is less prone to hallucinations than GPT3.5 [15], its increased costs contradicted our aim of creating an open-source model that could be used at scale, without high costs.

Table 1 displays the size of the models that were taken into consideration for this study. The FLAN small and base models were useful for fast initial experimentation, but they are not featured in the Section 3 as their small size does not provide the models enough expressiveness to perform well on these tasks. The earlier GPT-1 and GPT-2 models were not analyzed in this study as they were similar in size and performance to the smaller FLAN-T5 models. The GPT4 model was also not included, as the costs for using it were considerably larger than GPT3.5-turbo, given the wide range of experiments to be performed and our final aim to introduce an open-source model.

**Table 1.** FLAN and GPT model sizes.

| Name | Size |
| --- | --- |
| FLAN-T5 small | 60 M |
| FLAN-T5 base | 250 M |
| FLAN-T5 large | 780 M |
| FLAN-T5 XL | 3 B |
| FLAN-T5 XXL | 11 B |
| GPT-1 | 117 M |
| GPT-2 | 1.5 B |
| GPT-3 | 175 B |
| GPT3.5-turbo | 150 B–175 B |
| GPT4 | unknown |

### 1.2. Current Study Objective

The overarching objective of this study was to develop an automated model for evaluating the comprehension strategies (paraphrasing, elaboration, and bridging) employed by readers and the overall quality of the produced self-explanations. These tools can be useful in enhancing the capabilities of an Intelligent Tutoring System (ITS), designed to improve students' reading comprehension by having them practice reading and self-explaining in an environment in which timely feedback and evaluation are offered.

This study was focused on evaluating the extent to which open-source Large Language Models (LLMs) can be leveraged to build such an automated system. The results were compared to the performance of previous methods, which relied on smaller and less-resource-intensive machine learning models [7]. We also analyzed how the performance of these LLM models scaled with the model and prompt size. We provide a side-by-side comparison between open-source models and the OpenAI API used as the backbone of the popular ChatGPT. We released our best model on HuggingFace and the corresponding code on GitHub: https://github.com/readerbench/self-explanations, accessed on 13 October 2023.

The paper is structured into four sections following the Introduction. The Section 2 begins with a short description of the corpus on which the experiments were based. It then offers an overview of prompting for LLMs, describing a template for the prompts to be

used in the experiments. Lastly, it provides an overview of LLM fine-tuning methods. The Section 3 thoroughly analyzes the models' performance in both an "out-of-the-box" and fine-tuned setting. The Section 4 analyzes the performance of the best models, while the Section 5 reiterates the main findings and their importance.

## 2. Method

### 2.1. Corpus

The corpus used in this study consisted of three datasets containing 11,833 annotated self-explanations [26]. The datasets were collected from high school and undergraduate students who were asked to read one or two science texts and generate self-explanations for 9 to 16 target sentences. An entry consists of the target sentence, a self-explanation, and categorical scores for paraphrase presence, bridging, elaboration, and overall self-explanation quality. All entries were scored by pairs of expert readers in accordance with a common rubric [26].

The corpus was split into train/dev/test using a ratio of 54.5%/27.5%/18%. The categorical scores for the four tasks ranged from 0 to 2 or 3. The problem of predicting these scores was modeled as a classification task, with each score representing a class. The values were codified consistently across tasks so that Class 0 always represented low-quality self-explanations or the absence of a particular strategy. In contrast, higher values represented self-explanations of higher quality.

Class imbalance was an issue for each of the 4 tasks, as seen in Table 2. In the case of simple tasks, such as detecting the paraphrasing strategy's presence, the large majority of the samples contained high-quality paraphrasing (i.e., Class 2), with few examples for Classes 0 and 1. In the case of more-difficult tasks, such as assessing elaboration presence or self-explanation overall quality, the reverse happened. High-quality examples were in short supply, while low-quality samples (in the case of elaboration presence) or average-quality examples (in the case of overall SE quality) were more numerous. To reduce this imbalance, the final 2 classes for the bridging presence and elaboration presence tasks containing higher-quality examples were merged. After these changes, the elaboration presence task had 2 classes, paraphrase and bridge presence had 3 classes, whereas the overall quality task had 4 classes, as seen in Figure 1.

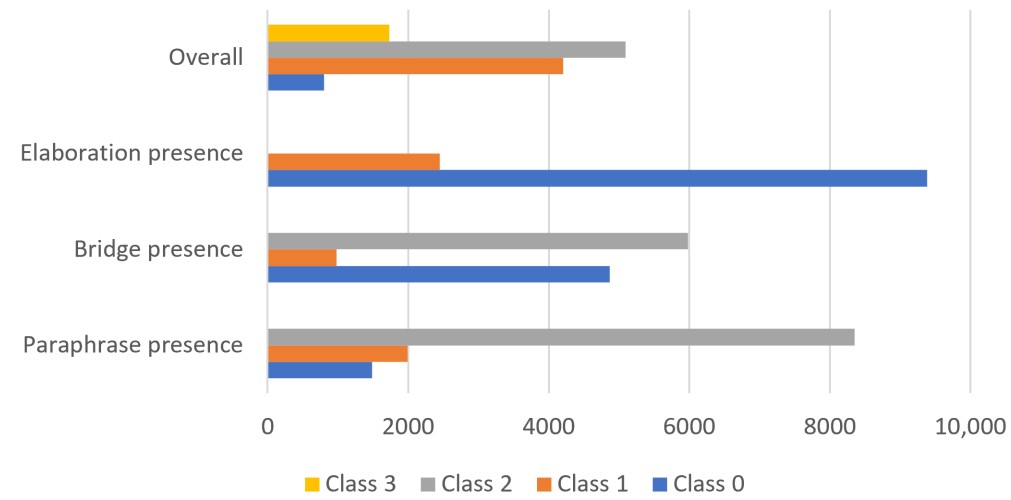

**Figure 1.** Class distribution per task after merging classes for elaboration and bridging.

**Table 2.** Class distribution per task.

| Dimension | Class 0 | Class 1 | Class 2 | Class 3 |
|---|---|---|---|---|
| Paraphrase presence | 1487 | 1992 | 8354 | |
| Bridge presence | 4869 | 981 | 4569 | 1414 |
| Elaboration presence | 9382 | 777 | 1674 | |
| Overall | 799 | 4207 | 5093 | 1734 |

### 2.2. LLM Prompting

The format of the prompt (i.e., input text for the LLM) can influence the quality of the provided answer [27]. Therefore, we tried to structure the input similarly to how the input was structured for the tasks on which the initial FLAN-T5 model was trained, as seen in the Section B Appendix of the original FLAN T5 paper [17]. That structure usually consists of a context, a set of examples of the task being solved, and the target task (see Figure 2). Additionally, we experimented with adding a "System role" entry at the beginning of the prompt for the requests made to GPT3.5-turbo, as suggested by the OpenAI GPT3.5-turbo API documentation [28]. This section provides a description of the perspective from which the model should approach the task (e.g., "You are a high school student tasked to summarize this text"). The "Context" section provides additional descriptive information regarding the task to be solved.

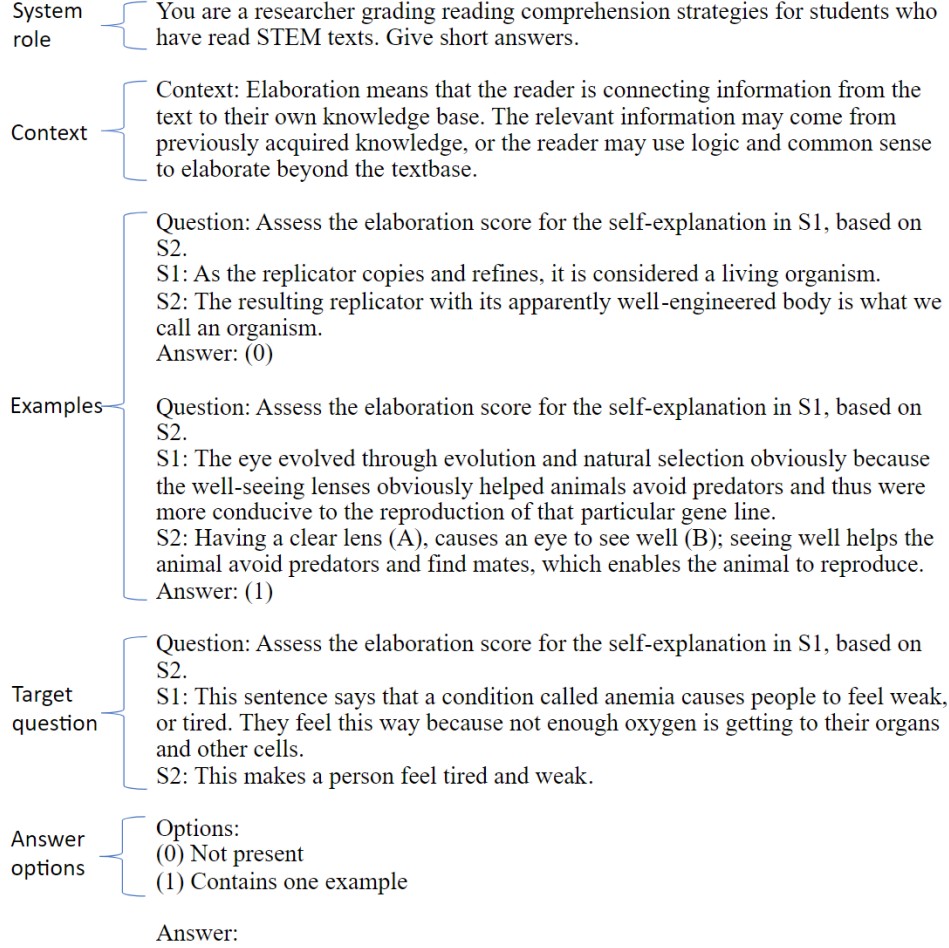

**Figure 2.** Example of prompt for the elaboration task in a multi-shot setting.

Both the FLAN-T5 models and the GPT3.5-turbo API were queried in 0-shot, 1-shot, and multi-shot settings to evaluate how examples can assist the model in providing better answers. In the multi-shot setting, the model was provided one example per class, selected from the training set. This was feasible because the tasks had a maximum of four classes.

The "Target question" section contains the question that the model must answer. Since answering the question involves reading the generated self-explanation and the source sentence, we added them to this section and labeled them as "S1" and "S2". Preliminary experiments indicated that the models performed better when using these naming conventions rather than "Generated Sentence" and "Original Sentence" or other combinations.

Lastly, the "Answer options" section lists the possible answers and a short description. Experiments were also performed with more-detailed descriptions of the classes, but this only improved performance in the case of the GPT3.5-turbo experiments.

### 2.3. LLM Fine-Tuning with LoRA

There are multiple methods of adapting pretrained LLMs to help them perform better on certain tasks. One such option is using the last set of hidden features that the model produces and training a small deep learning model to predict the expected output based on the set of hidden features while freezing the updates for the LLM parameters. This is efficient in terms of resources, but can add latency on inference because the depth of the model is increased. A second option consists of selective fine-tuning, in which only a subset of the LLM's layers are trained while the rest are kept unchanged. This approach can also be efficient, but it involves manually selecting which layers to train, an operation that is not necessarily intuitive. The third option consists of fine-tuning the entire model. Out of the three approaches, this should yield the best results, but it requires the most GPU memory and training resources.

Apart from the classical methods listed above, Parameter-Efficient Fine-Tuning (PEFT) methods rely on training only a small set of parameters without manually selecting what parameters to train. Some techniques, such as P-Tuning [29], focus on training a small encoder network for the prompt to produce a soft prompt based on the original prompt, relying on the assumption that better prompting can improve task performance. Prefix Tuning [30] takes that approach one step further by adding a small set of trainable parameters for each layer, which generates a soft prefix prompt concatenated to the actual prompt. Other techniques, such as Low-Rank Adaptation (LoRA) (i.e., Low-Rank Adaptation of Large Language Models, [31]), focus on a subset of parameters for every layer obtained via low-rank decomposition. Compared with P-Tuning and Prefix Training, LoRA has to train more parameters, but it is more suitable for adapting the LLM to a task different from the initial training tasks. The other two approaches are more suitable for adapting the LLM to a similar task from the ones on which it was pretrained.

LoRA efficiently fine-tunes LLMs by freezing the pretrained model and injecting trainable rank decomposition matrices into each layer. The authors claim that LoRA can reduce the number of trainable parameters by 10,000-times and the GPU memory requirement by 3-times for a 175-billion-parameter GPT-3 training. Furthermore, the method adds no extra inference latency.

The innovation that LoRA brings is the use of low-rank parametrized update matrices. In a classical fine-tuning setting for a weight matrix $W_0 \in \mathbb{R}^{d \times k}$, we would have an update after backpropagation equivalent with $W = W_0 + \Delta W$ with $\Delta W$ having the same dimensions as the pretrained matrix. LoRA considers the following decomposition: $\Delta W = BA$ with $B \in \mathbb{R}^{d \times r}, A \in \mathbb{R}^{r \times k}$, with rank $r \ll min(d, k)$. The two low-ranked matrices, A and B, will be trainable throughout the run while $W_0$ is frozen and initialized so that the initial update matrix is 0. As such, LoRA was the best alternative when fine-tuning the FLAN-T5 models.

## 3. Results

In this section, we explore the extent to which the performances presented in previous studies [7] can be surpassed by employing out-of-the-box or fine-tuned LLMs. The input received by the models consisted of prompts like the ones described in Section 2.2, which contained the student's self-explanation and the target sentence. Because of computational constraints, we skipped the scenarios in which the target sentence was omitted or was extended by including the previous sentences.

We used the F1-score as the evaluation metric for the results. Because LLMs can generate incorrectly formatted answers, we considered all badly formatted answers as belonging to Class 0, which have been coded to contain low-quality examples. The percentage of correctly formatted answers is also reported in order to understand how well the models have adapted to the task format.

We analyzed the percentage of correctly formed answers on the overall task to observe how well the models adapted to the task format. The FLAN-T5 large and GPT3.5-turbo models conformed to the expected format of answers (see Figure 3). Replies generated by the FLAN-T5 XL and XXL versions improved (i.e., followed the correct format) when they were presented with more examples in the prompt. When looking at the output of the models, we observed that the FLAN-T5 XL and XXL models tended to provide more verbose replies, not necessarily incorrect, but did not match the expected format.

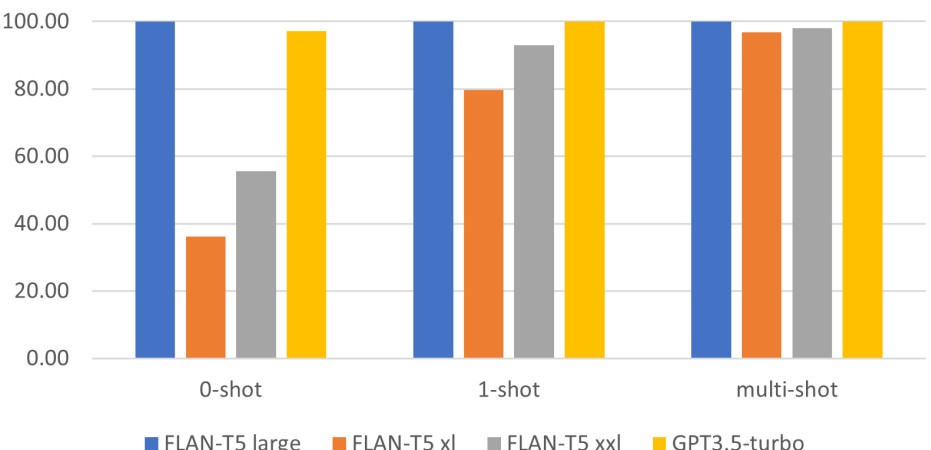

**Figure 3.** Percentage of correctly formed answers for the "out-of-the-box" evaluation.

The results were grouped into two sub-sections: the first subsection focuses on the "out-of-the-box" performance of FLAN-T5 and GPT3.5-turbo, while the second subsection presents the performance of FLAN-T5 after fine-tuning using the LoRA method.

### 3.1. Out-of-the-Box Performance

In this section, the "out-of-the-box" performance is assessed without fine-tuning. The models covered in this section are the FLAN-T5 large, XL, and XXL models, along with the GPT3.5-turbo API. The same prompt structure was used for all FLAN-T5 models. This structure was chosen after a series of experiments evaluating how small changes in the prompt can affect the model's performance. The final version of the prompt for FLAN-T5 resembled the structure described in Figure 2, with some modifications. It did not include a context section, whereas the prompt had shortened versions for the options in the Answer options section, labeled alphabetically instead of numerically (i.e., (A), (B), (C), (D), instead of (0), (1), (2), (3)). The prompt for the GPT3.5-turbo model had a context section and used long answer options labeled alphabetically.

The results for this task are presented in Table 3. For every task, the best result is listed using bold font. In some cases, multiple examples are bolded for a single task because the differences between the results were marginal.

**Table 3.** Out-of-the-box results for FLAN-T5 and GPT3.5-turbo.

| Task | Model | Weighted F1-Score | | |
|---|---|---|---|---|
| | | **0-Shot** | **1-Shot** | **Multi-Shot** |
| Paraphrase | FLAN-T5 large | **75.23%** | 66.30% | 68.73% |
| | FLAN-T5 XL | 28.24% | 53.93% | **75.26%** |
| | FLAN-T5 XXL | 0.69% | 1.53% | 1.65% |
| | GPT3.5-turbo | 2.19% | 14.39% | 24.74% |
| Elaboration | FLAN-T5 large | 50.09% | 58.78% | 56.94% |
| | FLAN-T5 XL | **89.90%** | **89.87%** | **89.90%** |
| | FLAN-T5 XXL | 87.79% | 78.52% | 80.11% |
| | GPT3.5-turbo | 44.53% | 55.38% | 56.13% |
| Bridging | FLAN-T5 large | 45.24% | 45.39% | 45.04% |
| | FLAN-T5 XL | 34.30% | **51.61%** | 44.34% |
| | FLAN-T5 XXL | 23.06% | 22.76% | 22.85% |
| | GPT3.5-turbo | 19.26% | 32.80% | 41.18% |
| Overall | FLAN-T5 large | 27.97% | 9.73% | 7.09% |
| | FLAN-T5 XL | 2.84% | 7.24% | 6.64% |
| | FLAN-T5 XXL | 10.68% | 8.81% | 12.44% |
| | GPT3.5-turbo | **30.18%** | 28.07% | 27.87% |

*Note.* Bold marks the best performance for every task.

The FLAN-based models performed considerably better than the GPT3.5-turbo model for the three comprehension strategy tasks. Differences of 51% (paraphrase presence), 33% (elaboration presence), and 10% (bridging presence), in terms of weighted F1-scores, were observed between the best FLAN-T5 performance and the best GPT3.5-turbo results.

The model size did not have a large influence on the performance in the case of the FLAN-T5 models. The differences between FLAN-T5 large, XL, and XXL were unclear. One possible explanation is that, as Figure 3 indicates, the XL and XXL models generated badly formatted responses because they tended to provide more verbose replies. This effect could have brought down the performance of these models, compared to the FLAN-T5 large model, which did not exhibit this phenomenon. The best results were obtained with a FLAN-T5 XL model for the bridging and elaboration presence tasks (see Table 3). The best performance on the paraphrase presence task was a tie between FLAN-T5 large and XL, while FLAN-T5 large performed the best for the overall task.

The impact on the performance of providing the model with more examples via the prompt was unclear for the FLAN-T5 models since no clear pattern can be observed in this regard. An improvement was seen regarding the percentage of correctly formed replies for the XL and XXL models, as seen in Figure 3. When exposed to multiple examples in the prompt, these models had a similar rate of correctly formed responses to GPT3.5-turbo and FLAN-T5 large. In the case of the GPT3.5-turbo model, the impact of adding examples was considerably clearer. We can observe that the results for the three comprehension strategy tasks improved when switching from 0-shot to 1-shot prompting and further on when switching to the multi-shot setting. However, the reverse happened for the overall quality task. The GPT3.5-turbo model performed worse as more examples were added.

Further exploration was undertaken for the prompting format used to query the GPT3.5-turbo API in the multi-shot scenario. The endpoint was queried using more examples, the context section, and extended descriptions of the classes. Adding more examples on top of the multi-shot setting did not help. However, adding the context and the extended descriptions slightly improved the results for some tasks. These prompt changes resulted in a slight improvement for the overall and bridging classes, considerable improvement for the paraphrase class, and a high drop in performance for the elaboration class. The results for the best-performing prompt are listed in Table 4. The results of these prompts are referenced when presenting the confusion matrices and the qualitative analysis for the GPT3.5-turbo model.

**Table 4.** GPT3.5-turbo performance after exploring prompt variations.

| Task | Model | Weighted F1-Score |
|------|-------|-------------------|
| Paraphrase | GPT3.5 | 65.54% |
| Elaboration | GPT3.5 | 6.80% |
| Bridging | GPT3.5 | 44.07% |
| Overall | GPT3.5 | 30.67% |

### 3.2. Fine-Tuning

In this section, we analyze the performance of fine-tuning the FLAN-T5 models using the LoRA method. Experiments were run using the publicly available FLAN-T5 models on HuggingFace; similarly, the FLAN-T5 small and base versions were excluded in this subsequent analysis, given their poor performance.

Three FLAN-T5 models were initially trained using a small learning rate of $3 \times 10^{-4}$ for one epoch (i.e., one pass through the entire dataset) on the four tasks in the 0-shot, 1-shot, and multi-shot settings. The learning rate was chosen after running multiple experiments and observing the evolution of the training loss and the final performance of the model. Models trained with larger learning rates converged faster with worse outcomes, while models trained with smaller learning rates did not always converge in one epoch. All models were trained using a mini-batch size of 1. The FLAN-T5 XL and XXL models were constrained to do so by the limited amount of GPU memory. In the case of the smaller models, experiments with larger mini-batch sizes were faster, but led to poorer results, probably because the learning rate had to be adapted depending on the batch size [32]. No other hyper-parameter was tuned besides the learning rate and the mini-batch size. The same prompt structure as in the "out-of-the-box" scenario was considered.

The performance for the paraphrase, bridging, and overall quality tasks improved considerably when switching from 0-shot to 1-shot and then to multi-shot settings, as seen in Table 5. In the case of elaboration presence, the pattern was not as clear, but the best result was still obtained in a multi-shot setting. One exception was the performance of the FLAN-T5 XXL model on the overall task, which required fine-tuning the learning rate to achieve a good performance. The standard learning rate for the fine-tuning experiments was $3 \times 10^{-4}$, but this model obtained its best performance using $1.5 \times 10^{-4}$; most likely, the XXL model is more sensitive to the gradient update step size, and a larger learning rate would cause it to oscillate around a narrow local minimum, without reaching it.

When looking at the impact of model size on performance, larger models tended to perform better for all the tasks. The best result for every task was obtained using the FLAN-T5 XXL model, and we can also observe that the FLAN-T5 XL outperformed the large variant in most scenarios.

**Table 5.** One-epoch fine-tuned results for FLAN-T5.

| Task | Model | Weighted F1-Score | | |
|------|-------|--------|--------|------------|
| | | 0-Shot | 1-Shot | Multi-Shot |
| Paraphrase | FLAN-T5 large | 21.45% | 68.46% | 82.53% |
| | FLAN-T5 XL | 10.63% | 37.54% | 85.50% |
| | FLAN-T5 XXL | 72.79% | 74.98% | **86.76%** |
| Elaboration | FLAN-T5 large | 83.99% | 84.12% | 74.26% |
| | FLAN-T5 XL | 87.66% | 81.58% | 84.28% |
| | FLAN-T5 XXL | 88.64% | 88.63% | **89.80%** |
| Bridging | FLAN-T5 large | 42.68% | 45.32% | 45.85% |
| | FLAN-T5 XL | 24.37% | 48.22% | 61.26% |
| | FLAN-T5 XXL | 53.13% | 76.32% | **79.06%** |
| Overall | FLAN-T5 large | 1.34% | 2.15% | 36.22% |
| | FLAN-T5 XL | 11.53% | 25.70% | 40.02% |
| | FLAN-T5 XXL | 59.68% | **64.39%** | 61.25% [1] |

*Note.* Bold marks the best performance for every task. [1] Obtained after extra hyper-parameter tuning.

As previously mentioned, the models were evaluated in a scenario where badly formatted answers were labeled as low-quality, Class 0. For this reason, it is also important to consider the percentage of correctly formed answers. Figure 4 shows that the percentage of correctly formed answers increased as more examples were added to the prompt. The results for FLAN-T5 large were dramatically low in the 0-shot and 1-shot settings, but they considerably improved for multi-shot. The same trend is visible for the FLAN-T5 XL and XXL models. We can also observe that the larger models tended to better format the answers correctly.

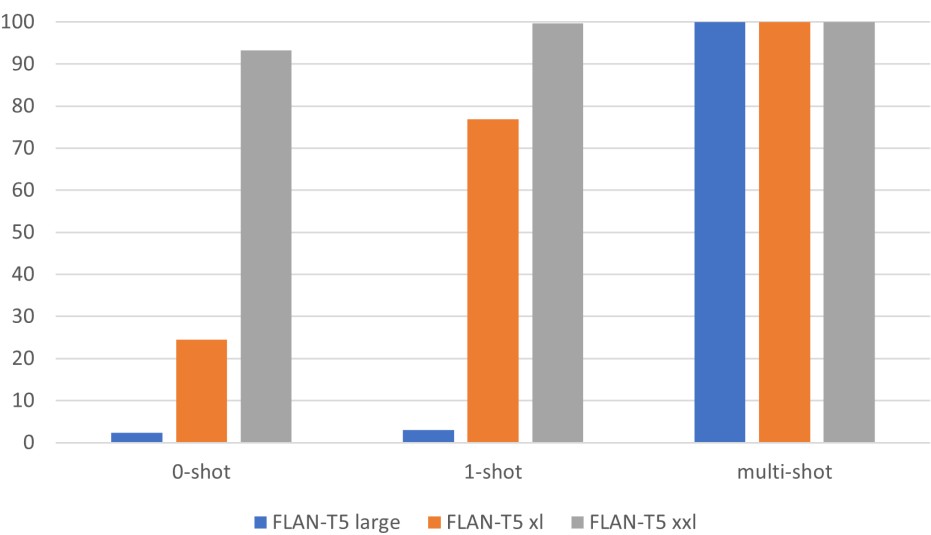

**Figure 4.** Percentage of correctly formed answers for fine-tuned models on the overall task.

Lastly, experiments were performed to observe whether model performance improved if fine-tuning for more epochs (see Table 6). Preliminary experiments indicated that the test loss would reach a plateau after three epochs of fine-tuning. In order to reduce the number of experiments, we evaluated the FLAN-T5 large, XL, and XXL models only in the multi-shot setting.

**Table 6.** Three-epoch fine-tuned results for FLAN-T5.

| Task | Model | Scenario | Weighted F1-Score |
|---|---|---|---|
| Paraphrase | FLAN-T5 large | multi-shot | 86.70% |
| | FLAN-T5 XL | multi-shot | **86.76%** |
| | FLAN-T5 XXL | multi-shot | 86.21% |
| Elaboration | FLAN-T5 large | multi-shot | 89.33% |
| | FLAN-T5 XL | multi-shot | **89.88%** |
| | FLAN-T5 XXL | multi-shot | 89.54% |
| Bridging | FLAN-T5 large | multi-shot | 63.72% |
| | FLAN-T5 XL | multi-shot | **79.02%** |
| | FLAN-T5 XXL | multi-shot | **79.02%** |
| Overall | FLAN-T5 large | multi-shot | 58.49% |
| | FLAN-T5 XL | multi-shot | 69.85% |
| | FLAN-T5 XXL | multi-shot | **72.12%** |

*Note.* Bold marks the best performance for every task.

In this scenario, FLAN-T5 XL performed better in 2 out of 4 cases, while FLAN-T5 XXL considerably outperformed the other two on the overall task. The FLAN-T5 large model obtained good results on the paraphrase and elaboration tasks, but had worse results on the remaining two. For the three comprehension strategy tasks, the results were close when comparing the XL and XXL models, with the XL model having a slight advantage. It must be noted that, because the FLAN-T5 XXL model was the best-performing model,

extra hyper-parameter tuning was performed to maximize its potential. In the end, this model was trained for all tasks using a smaller learning rate of $1.5 \times 10^{-4}$, as opposed to the standard $3 \times 10^{-4}$ used for the other models.

The fine-tuned FLAN-T5 XXL model obtained the best performance on the overall task, surpassing even the results from previous work (see [7]). The best models in that study were Single-Task (STL) and Multi-Task (MTL) neural network architectures based on a pretrained RoBERTa model. The LLM-based methods obtained a better result for the overall, paraphrase, and bridging presence tasks, while the MTL/STL models still held a narrow edge over them on the elaboration presence task (see Table 7).

**Table 7.** Best results across the two studies.

| Task | Previous Results [7] | | Current Study | | |
|---|---|---|---|---|---|
| | **Best Model** | **Scenario** | **Best Model** | **Scenario** | **Improvement** |
| Paraphrase | 84.30% | STL | 86.76% | Fine-tuned XXL multi-shot | 2.46% |
| Elaboration | 89.90% | STL | 89.88% | Pretrained XL | −0.02% |
| Bridging | 78.50% | STL | 79.02% | Fine-tuned XXL multi-shot | 0.52% |
| Overall | 69.90% | MTL | 72.12% | Fine-tuned XXL multi-shot | 2.12% |

## 4. Discussion

This study evaluated the performance of LLMs on scoring self-explanations using multiple employed strategies in either out-of-the-box or fine-tuned setups. In the out-of-the-box scenario, a comparison was made between the performance of the FLAN-T5 models and the GPT3.5-turbo API. The FLAN-T5 models obtained better results on three comprehension strategy tasks. The model performance did not scale with the model size and the number of examples listed in the prompts. The GPT3.5-turbo model obtained better results on the overall quality task and showed a clearer improvement on the other tasks with the addition of more examples to the prompt.

When analyzing the correctness of the responses generated by the LLMs, it was also observed that GPT3.5-turbo and FLAN-T5 large were more likely to generate answers in the correct format. This capability improved for all the models if more examples were provided in the prompt.

**Table 8.** Confusion matrices for the "out-of-the-box" models on the overall task.

| FLAN-T5 Large 0-Shot | Predicted 0 | Predicted 1 | Predicted 2 | Predicted 3 |
|---|---|---|---|---|
| Actual 0 | 28 | 7 | 14 | 12 |
| Actual 1 | 283 | 254 | 30 | 121 |
| Actual 2 | 428 | 130 | 60 | 308 |
| Actual 3 | 182 | 21 | 54 | 210 |
| **GPT3.5 Multi-Shot** | **Predicted 0** | **Predicted 1** | **Predicted 2** | **Predicted 3** |
| Actual 0 | 18 | 10 | 13 | 20 |
| Actual 1 | 274 | 107 | 109 | 198 |
| Actual 2 | 207 | 134 | 222 | 363 |
| Actual 3 | 47 | 27 | 104 | 289 |

When looking at the confusion matrix for the overall task, the two best-performing out-of-the-box models tended to misclassify multiple examples, not only in adjacent classes, but in other classes as well (see Table 8). Numerous instances of Class 0 examples were classified as Class 3 and vice versa. This indicated that the models could not reliably identify content that had been copied and pasted. There were even more high-quality examples, namely Class 3, being labeled as low-quality, or Class 0. This could indicate that the models have not completely understood the task. They might be solving a proxy task, such as paraphrase assessment, with similar scores in some cases and diverging scores in others. For instance, a good self-explanation might contain relevant paraphrases; however, good self-explanations should target information beyond the source text. In addition, the

predictions can also be influenced by high class imbalance (i.e., Class 0 had almost nine times fewer examples than Class 2 for self-explanation quality).

In the fine-tuning scenario, only the FLAN-T5 models were targeted. Initially, the models were fine-tuned for one epoch using the LoRA method. After this fine-tuning, the performances drastically improved and scaled better with the model size and number of examples provided. When the models were trained for three epochs, the differences between the FLAN-T5 XL and XXL models decreased.

**Table 9.** Confusion matrices for the best fine-tuned FLAN-T5 model on the overall task.

|  | **Predicted 0** | **Predicted 1** | **Predicted 2** | **Predicted 3** |
| --- | --- | --- | --- | --- |
| Actual 0 | 18 | 21 | 18 | 4 |
| Actual 1 | 39 | 512 | 132 | 5 |
| Actual 2 | 1 | 110 | 735 | 80 |
| Actual 3 | 0 | 5 | 179 | 283 |

The confusion matrix generated for the best-performing, fine-tuned model on the overall task showed improved results compared to the out-of-the-box models (see Table 9). We observed that most predictions coincided with the ground truth for almost all classes, except the underrepresented Class 0. Furthermore, even when errors occurred, they appeared near the correct options; only four instances occurred at a distance from three classes (i.e., Class 0 examples evaluated as Class 3). Furthermore, there was no example of a high-quality sample (i.e., Class 3) being labeled as a poor-quality self-explanation (i.e., Class 0).

The FLAN fine-tuned models and the previous MTL approach can also be compared in regards to the training time, as reported in Table 10. We can observe that the MTL model required the least training time while using less-performant hardware. For the FLAN models, the training time listed was for 1 epoch, so the 3-epoch fine-tuned model would take roughly three times more time to train. Our best-performing three-epoch-trained FLAN XXL surpassed the previous MTL model performance, but that model required 540 min to train (a 27× increase) and more expensive hardware.

**Table 10.** Training time per model.

| **Model** | **No. of Epochs** | **Total Training Time (Minutes)** | **GPU Type** |
| --- | --- | --- | --- |
| MTL | 25 | 20 | Tesla P100 |
| FLAN large | 1 | 23 | Tesla P100 |
| FLAN XL | 1 | 100 | Tesla A100 40 GB |
| FLAN XXL | 1 | 180 | Tesla A100 40 GB |

*4.1. Error Analysis*

Table 11 lists 10 randomly selected inputs, at least one per class, on which the following three models were evaluated: the best-performing model (FLAN-T5 XXL multi-shot), the best-performing out-of-the-box FLAN-T5 model (FLAN-T5 large 0-shot), and GPT3.5-turbo (prompted in a multi-shot setting). Our evaluation parser considered all listed outputs valid, despite GPT3.5's extra verbosity in listing the class description along with the class name or the lack of parenthesis for FLAN-T5 large on Example 1. The models performed better when prompted with alphabetical classes as options instead of numerical ones. For this reason, the classes appear with a different naming convention in this table compared to previous mentions. However, the correspondence is easy to understand as Class (A) corresponds to Class 0, (B) to Class 1, (C) to Class 2, and (D) to Class 3.

Examples 2 and 3 showed all models answering correctly when classifying input belonging to Classes 2 and 3. There were also cases (Examples 4 and 5) of minor errors, where the models classified a good example as having high quality. One possible explanation is that the self-explanations were particularly verbose, and the models had trouble keeping track of all the information and comparing it with the source text.

Both out-of-the-box models had an example three classes away from the ground truth (see Example 6). This is a classic example of copy-pasted content, labeled as low-quality, because the reader did not make an effort to self-explain the source text. The fine-tuned FLAN-T5 XXL model detected this and correctly rated the example as poor quality.

The performance obtained by these models on this 10-example subset was consistent with the previously presented results. The best-performing FLAN-T5 XXL model was correct in 9 out of 10 situations and was one class away from the correct answer in the erroneous case. The out-of-the-box models managed to correctly answer in 2 or 3 out of 10 cases, exhibited errors 2 classes or 3 classes away from the ground truth, and had minor issues correctly formatting the output.

**Table 11.** Sample outputs.

| ID | Self-Explanation | Source Sentence | FLAN-T5 XXL Multi-Shot | FLAN-T5 Large 0-Shot | GPT3.5-Turbo | Ground Truth |
|---|---|---|---|---|---|---|
| 1 | This sentence explains that the circular shape of the red blood cells result in a big surface area, which lets them be efficient at gas diffusion. | The disk shape of red blood cells results in a large surface area, which enables them to be efficient at gas diffusion | (B) | C | (C) Good quality | (B) |
| 2 | Red blood cells gets its color from the Hemoglobin. | Hemoglobin also contains iron, which gives blood its red color | (B) | (B) | (B) Fair quality | (B) |
| 3 | This sentence explains how hemoglobin, a complex protein in red blood cells, binds to the oxygen and carbon dioxide that the red blood cells transport. | Hemoglobin binds to the oxygen and carbon dioxide that the red blood cells transport | (C) | (C) | (C) Good quality | (C) |
| 4 | This sentence is saying that red blood cells have essentially two jobs, the second of which being the removal of carbon dioxide that is no longer needed. Oxygen enters the body, and waste carbon dioxide leaves the body with the help of red blood cells. | They also pick up waste carbon dioxide for removal | (C) | (C) | (C) Good quality | (D) |
| 5 | Red blood cells carry oxygen to the cells and remove waste. The way they are shaped allows gas diffusion to go well. Once the red blood cells have the oxygen and carbon dioxide waste, hemoglobin binds them. | Hemoglobin binds to the oxygen and carbon dioxide that the red blood cells transport | (D) | (C) | (C) Good quality | (D) |
| 6 | the shape of the cells causes them to clog blood vessels, preventing oxygen from reaching muscles and other tissues | The shape of the cells causes them to clog blood vessels, preventing oxygen from reaching muscles and other tissues | (A) | (D) | (D) High quality | (A) |
| 7 | When low amounts of oxygen are transported, a person can feel tired or weak due to the body not being replenished completely.The heart, lungs, and muscles rely on oxygen to function, so if there is a deficiency of that a person would become fatigue. | This makes a person feel tired and weak | (D) | (C) | (B) Fair quality | (D) |
| 8 | if you have a lot of iron, it will make your blood red | Hemoglobin also contains iron, which gives blood its red color | (B) | (C) | (A) Poor | (B) |
| 9 | This means that because of the red blood cells shape being like a disk it helps the body with gas diffusion. Like if the body has a lot of gas build up in it then the red blood cells help get rid of the gas. | The disk shape of red blood cells results in a large surface area, which enables them to be efficient at gas diffusion | (C) | (C) | (B) Fair quality | (C) |
| 10 | As a result, the person feels sluggish and has less energy. They are lacking the oxygen which presumably messes up their oxygen:carbon dioxide ratio. | This makes a person feel tired and weak | (C) | (A) | (B) Fair quality | (C) |

*4.2. Limitations*

The class imbalance was an impediment in some cases, especially for the elaboration presence task, where Class 0 accounted for roughly 93% of samples. This made it tempting for the fine-tuned models to disproportionately label new examples as low quality since it seemed like a sure bet. For that reason, more focus in the analysis was put on the overall quality task, which was also one of the more balanced tasks in addition to being more complex.

Especially in regard to the GPT models, our experiments were restricted by the costs of using the API. Because we sought to explore multiple scenarios (i.e., 0-shot, 1-shot, multi-shot) for all four tasks and also explore different variations of prompts, doing so for multiple OpenAI endpoints would have increased our cost, especially because the GPT4 model is roughly 20–30× more expensive than GPT3.5-turbo.

The experiments presented in this study were not evaluated in an iterative setting, where the request is not modeled as a monolithic prompt, but as a dialogue with multiple short requests. The initial requests could have provided the context and the examples, while the last request could have focused solely on the classification task. This would have been more advantageous for the GPT3.5-turbo model, which is targeted more towards usage in a conversational setting.

Lastly, a limitation of the currently proposed methods is the fact that they have high resource requirements. Even for the "out-of-the-box" setting, the FLAN-T5 XXL model could only be deployed on an NVidia A100 GPU because its 11-billion parameters are stored in floating point precision. Having such a large model permanently deployed for a backend system that would serve the requests made by an intelligent tutoring system would be costly. In that regard, solutions for reducing the network's footprint could be used. These include either pruning channels [33] or entire layers [34], quantizing the weights and activations (i.e., reducing their precision) [35,36], or using a combination of both [37]. These approaches have also been applied in the case of LLMs, with methods such as LLM-Pruner [38], which performs a selective removal of nonessential structures in the network based on gradient information.

## 5. Conclusions and Future Work

This study, corroborated by the previous work of Nicula et al. [7], indicated that evaluating reading strategies and assessing overall self-explanation quality can be effectively addressed using deep learning models. This work showed that, with fine-tuning, pretrained LLMs surpass the performance of more-specialized medium-sized neural network architectures. The LLM models require a more expensive hardware setup for fine-tuning and can have more inference latency than shallower medium-sized models; still, they are easier to adapt to a new task than a specialized medium-sized model.

The experiments also illustrated how well these models can be fine-tuned on a small dataset (i.e., thousands of examples). The models performed well despite the slightly heterogeneous corpus, which was compiled from three datasets, one containing text generated by high-school students and two containing text produced by undergraduates. However, the topic of the target texts was narrow, as only STEM texts were targeted. These types of texts contain objective information presented concisely, making them a better target for this type of evaluation. Analyzing the performance of such models on data targeting other topics, with different characteristics (e.g., texts that have a degree of subjectivity), or data generated by another category of students (e.g., primary school students) would be interesting. However, the effort required to collect and label such data in similar amounts (i.e., thousands of samples) should not be underestimated.

These approaches can be leveraged to develop systems that evaluate readers' existing text comprehension abilities or gradually guide them to improve their performance. The models pretrained as part of this work can be integrated within a more complex system that can provide a set of Application Programming Interfaces (APIs) queried by an automated tutoring system, enabling it to provide timely feedback and evaluations to students.

A promising direction of research would be to leverage the existing datasets and infrastructure for fine-tuning LLMs to train text-to-text models that generate a self-explanation of a specified quality or that target a particular combination of reading strategies. The output of such models can be used in an educational setting as practice by requiring students to rate and label computer-generated self-explanations or by providing the students with examples of how certain strategies can be handled.

In essence, there are any number of possibilities and opportunities to pursue in leveraging LLMs and other advanced technologies to prompt students to engage in various strategies such as self-explanation, which offer strong promise to enhance deep comprehension, problem-solving, and critical thinking.

**Author Contributions:** Conceptualization, M.D. and D.S.M.; methodology, B.N., M.D. and D.S.M.; software, B.N.; validation, B.N., M.D. and D.S.M.; formal analysis, M.D.; investigation, D.S.M.; resources, R.B. and T.A.; data curation, R.B. and T.A.; writing—original draft preparation, B.N.; writing—review and editing, M.D., R.B., T.A. and D.S.M.; visualization, B.N.; supervision, M.D. and D.S.M.; project administration, M.D. and D.S.M.; funding acquisition, M.D. and D.S.M. All authors have read and agreed to the published version of the manuscript.

**Funding:** This work was supported by the Ministry of Research, Innovation, and Digitalization, project CloudPrecis, Contract Number 344/390020/06.09.2021, MySMIS code: 124812, within POC, the Ministry of European Investments and Projects, POCU 2014-2020 project, Contract Number 62461/03.06.2022, MySMIS code: 153735, the IES U.S. Department of Education (R305A130124, R305A190063, the Office of Naval Research (N00014-20-1-2623), and the National Science Foundation (NSF REC0241144; IIS-0735682). The opinions expressed are those of the authors and do not represent the views of the Institute, the U.S. Department of Education, ONR, or NSF.

**Institutional Review Board Statement:** The study was conducted according to the guidelines of the Declaration of Helsinki and approved by the Institutional Review Board of Arizona State University.

**Informed Consent Statement:** Informed consent was obtained from all subjects involved in the study.

**Data Availability Statement:** The code, as well as the links to models published on HuggingFace can be found at https://github.com/readerbench/self-explanations (accessed on 12 September 2023).

**Conflicts of Interest:** The authors declare no conflict of interest.

## Abbreviations

The following abbreviations are used in this manuscript:

| | |
|---|---|
| API | Application Programming Interface |
| BERT | Bidirectional Encoder Representations from Transformers |
| CoT | Chain-of-Thought |
| LLM | Large Language Model |
| LoRA | Low-Rank Adaptation |
| NLP | Natural Language Processing |
| SE | Self-Explanation |
| SERT | Self-Explanation Reading Training |
| STEM | Science, Technology, Engineering, and Mathematics |

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
