# Peer review of "Automated Assessment of Comprehension Strategies from Self-Explanations Using LLMs"

_information, doi:10.3390/info14100567_

Round 1
Reviewer 1 Report
The paper presents a study of the use of the FLAN-T5 language model for the evaluation of self-explanations written in natural language. The authors present and thoroughly compare the results obtained with different variants of the FLAN-T5 model for 0-shot, 1-shot and multi-shot settings, and they also provide a comparison with the GPT3.5-turbo model. The whole paper is well written and understandable, the methods used and the results obtained are clearly described, evaluated and discussed. The authors provide a sufficient description of the related language models and their fine-tuning. The only weakness, in my opinion, is the rather short description of state-of-the-art approaches; the results are mostly compared to a single paper [7]. On the other hand, the whole research area is quite new. I appreciate that the code used for the experiments is available, which makes it possible to reproduce the results.
I have one small doubt: Figure 2 shows an example of a prompt. However, I couldn't find any reference to this figure in the text, even though section 2 seems to explain it. On the other hand, section 3.1 introduces some modifications of the prompt, so it seems that the prompt from Figure 2 was not used after all? I think it should be made clearer which prompts were actually used for which models.
In conclusion, I find the paper very nicely written and almost ready for publication with only a few minor issues I have listed above.
I have found a few minor formal issues:
- l. 7 (abstract): The sentence "Results improved..." does not make sense to me, some words are missing?
- l. 172 missing reference [?]
- l. 221 "do not _positively_ influence the results" - is this correct?
Author Response
The paper presents a study of the use of the FLAN-T5 language model for the evaluation of self-explanations written in natural language. The authors present and thoroughly compare the results obtained with different variants of the FLAN-T5 model for 0-shot, 1-shot and multi-shot settings, and they also provide a comparison with the GPT3.5-turbo model. The whole paper is well written and understandable, the methods used and the results obtained are clearly described, evaluated and discussed. The authors provide a sufficient description of the related language models and their fine-tuning. The only weakness, in my opinion, is the rather short description of state-of-the-art approaches; the results are mostly compared to a single paper [7]. On the other hand, the whole research area is quite new. I appreciate that the code used for the experiments is available, which makes it possible to reproduce the results.
Response: We did not have more examples of models trained on this corpus or on similar datasets. We tried to compensate by providing a more thorough overview of Large Language Models.
I have one small doubt: Figure 2 shows an example of a prompt. However, I couldn't find any reference to this figure in the text, even though section 2 seems to explain it. On the other hand, section 3.1 introduces some modifications of the prompt, so it seems that the prompt from Figure 2 was not used after all? I think it should be made clearer which prompts were actually used for which models.
Response: Thank you for the observation. Figure 2 represents a template for prompts that has been used with small changes in our experiments (e.g., the FLAN-T5 models did not use a System Role section, and they performed better without the context section). We have added references to Figure 2 in the text, where relevant.
In conclusion, I find the paper very nicely written and almost ready for publication with only a few minor issues I have listed above.
Response: We appreciate the feedback. Thank you kindly.
I have found a few minor formal issues:
- l. 7 (abstract): The sentence "Results improved..." does not make sense to me, some words are missing?
- l. 172 missing reference [?]
- l. 221 "do not _positively_ influence the results" - is this correct?
Response: Thank you for noticing these 3 issues. The sentence has been rephrased, the missing reference was corrected, and the statement at line 221 has been eliminated.
Reviewer 2 Report
This study is focused on leveraging open-source Large Language Models, i. e. FLAN-T5, to automatically assess the comprehension strategies employed by readers while understanding STEM texts. The results are compared to the performance of previous methods, which relied on smaller and less resource-intensive machine learning models.
The presentation is concise and well organized. The proposed methods and results are clearly presented. This paper should be accepted subject to the following conditions:
- Each acronym used in the paper have to be explained. For example, STEM acronym is not explained in the abstract.
- The organisation of the paper has to be mentioned in the end of Introduction.
- The contribution of the paper should be discussed with more details.
- The plans for future work have to be mentioned in the Conclusion.
Author Response
This study is focused on leveraging open-source Large Language Models, i. e. FLAN-T5, to automatically assess the comprehension strategies employed by readers while understanding STEM texts. The results are compared to the performance of previous methods, which relied on smaller and less resource-intensive machine learning models.
The presentation is concise and well organized. The proposed methods and results are clearly presented. This paper should be accepted subject to the following conditions:
- Each acronym used in the paper have to be explained. For example, STEM acronym is not explained in the abstract.
Response: Thank you for the observation, we have added explanations to the acronyms that were missing.
- The organisation of the paper has to be mentioned in the end of Introduction.
Response: We have added a paragraph describing the paper structure at the end of the Introduction section.
- The contribution of the paper should be discussed with more details.
Response: We have expanded the Conclusions section by providing more details on the contribution of the paper.
- The plans for future work have to be mentioned in the Conclusion.
Response: A paragraph regarding Future work, with focus on generative approaches, has been added at the end of the “Conclusions and Future Work” section.
Reviewer 3 Report
The study offers valuable insights into the potential of LLMs in evaluating comprehension strategies. Addressing the following points will enhance the clarity, depth, and relevance of the research, making it a significant contribution to the field. Looking forward to the revised manuscript.
Major Points for Revision:
-
Introduction:
- Contextual Background: The introduction should provide a clearer contextual background that highlights the current challenges or gaps in the field of automated assessment of comprehension strategies. This will help readers understand the study's significance in the broader context.
- Strategy Descriptions: The descriptions of various reading comprehension strategies need more clarity. Providing concise definitions or examples for each strategy will enhance understanding.
-
Large Language Models:
- Focus on LLMs: Reduce the emphasis on chatbots and provide a broader perspective on the applications and significance of LLMs, especially in the context of the study's objective.
- Model Selection Bias: Address potential biases or limitations arising from excluding certain models due to cost considerations.
-
Method - Corpus:
- Dataset Transparency: Provide more details about the nature, origins, and potential biases of the datasets used in the study.
- Class Imbalance: Offer detailed statistics about the initial class distributions and the methodology used to address class imbalance.
-
LLM Prompting:
- Prompt Structure Justification: Provide a clear rationale for the chosen prompt structures and naming conventions.
- Consistency in Prompting: Ensure consistency in the methodology, especially regarding the use of the "System role" entry for GPT3.5-turbo.
-
LLM Fine-tuning with LoRA:
- Fine-tuning Method Choice: Provide a clear justification for choosing LoRA over other fine-tuning methods.
- Hyperparameter Details: Offer a comprehensive account of the hyperparameter tuning process, including the rationale behind chosen learning rates.
-
Discussion:
- Model Performance Ambiguity: Delve deeper into why model performance did not scale with model size and the number of examples.
- Misclassification Analysis: Provide a comprehensive analysis of why models misclassify certain examples, especially between classes 0 and 3.
-
Conclusions:
- Generalizability: Address the generalizability of the study's findings to other contexts, texts, or age groups.
- Practical Implications: Discuss the real-world applications and potential challenges of implementing the models in educational settings.
Minor Recommendations:
- Consider citing the following works to enhance the depth and relevance of your literature review and discussions:
- doi.org/10.3390/bdcc6040130
- doi.org/10.1016/j.eswa.2022.119391
The quality of the English language used in the manuscript is generally good, with clear sentence structures and appropriate vocabulary. However, there are occasional instances where phrasing can be refined for better clarity and coherence.
Author Response
The study offers valuable insights into the potential of LLMs in evaluating comprehension strategies. Addressing the following points will enhance the clarity, depth, and relevance of the research, making it a significant contribution to the field. Looking forward to the revised manuscript.
Response: Thank you kindly for your thorough review.
Major Points for Revision:
- Introduction:
- Contextual Background: The introduction should provide a clearer contextual background that highlights the current challenges or gaps in the field of automated assessment of comprehension strategies. This will help readers understand the study's significance in the broader context.
Response: We appreciate the comment and understand the need for clarity. The field is narrow and we could not find similar tasks and models with which to compare our approaches. We tried to make our description of the reading strategies clearer. Furthermore, we also improved and added to the Conclusions section to mention how the techniques can be used to improve the results of Intelligent Tutoring Systems (ITS) like the ones referenced in the fifth paragraph of the Introduction section.
- Strategy Descriptions: The descriptions of various reading comprehension strategies need more clarity. Providing concise definitions or examples for each strategy will enhance understanding.
Response: We have restructured and improved the information in the introduction section to better highlight the 3 strategies (paraphrasing, bridging and elaboration).
- Large Language Models:
- Focus on LLMs: Reduce the emphasis on chatbots and provide a broader perspective on the applications and significance of LLMs, especially in the context of the study's objective.
Response: Thank you for the observation, we shifted the focus of the subsection from chatbots to LLMs, and added more details related to the GPT family of models.
- Model Selection Bias: Address potential biases or limitations arising from excluding certain models due to cost considerations.
Response: We added a comment in the GPT section acknowledging the increased performance of the GPT4 model, but also underlining the fact that its increased costs are in conflict with our aim of developing open-source tools to be integrated into Intelligent Tutoring Systems.
- Method - Corpus:
- Dataset Transparency: Provide more details about the nature, origins, and potential biases of the datasets used in the study.
Response: We have added more information related to the structure of the dataset and the way it was obtained.
- Class Imbalance: Offer detailed statistics about the initial class distributions and the methodology used to address class imbalance.
Response: A table outlining the class distribution for each task was added. We also commented on the natural class imbalance that occurs in some tasks, depending on their difficulty.
- LLM Prompting:
- Prompt Structure Justification: Provide a clear rationale for the chosen prompt structures and naming conventions.
Response: We have added information to the Prompting section for more clarity by referring to the section of the FLAN-T5 paper that contained prompt examples, and referencing Figure 2, which represented a prompt template, in the text, when relevant.
- Consistency in Prompting: Ensure consistency in the methodology, especially regarding the use of the "System role" entry for GPT3.5-turbo.
Response: We have clarified the meaning of the “System role” section, used for the GPT3.5-turbo prompts.
- LLM Fine-tuning with LoRA:
- Fine-tuning Method Choice: Provide a clear justification for choosing LoRA over other fine-tuning methods.
Response: Thank you for the suggestion. We have added a new paragraph focusing on multiple PEFT techniques (P-Tuning, Prefix Tuning and LoRA). We have also provided details about the motivation for choosing LoRA over the other two methods.
- Hyperparameter Details: Offer a comprehensive account of the hyperparameter tuning process, including the rationale behind chosen learning rates.
Response: We have provided additional details regarding our choice of 3e-4 learning rate and batch_size of 1. We also added an explanation as to why the 1.5e-4 learning rate was more suitable for the FLAN-T5 XXL models.
- Discussion:
- Model Performance Ambiguity: Delve deeper into why model performance did not scale with model size and the number of examples.
Response: We have added an explanation for why scaling was not visible in the case of the “out-of-the-box” experiments.
- Misclassification Analysis: Provide a comprehensive analysis of why models misclassify certain examples, especially between classes 0 and 3.
Response: Thank you for the observation, while we had an explanation for why class 0 was misclassified as class 3, the reverse case was not observed. We added an explanation for why the models could mistake a class 3 example (high quality) for a class 0 example (low quality). The explanation was that the models might be assessing a proxy task and not self-explanation quality. For that proxy task the examples could have been low quality, despite being high quality self-explanations.
- Conclusions:
- Generalizability: Address the generalizability of the study's findings to other contexts, texts, or age groups.
Response: We have extended the Conclusions section and added a paragraph focusing on the nature of the data, and the possibility of doing these analyses on data targeting other topics, or generated by a different category of students.
- Practical Implications: Discuss the real-world applications and potential challenges of implementing the models in educational settings.
Response: We have extended the Conclusions section to also include a description of how these models can be leveraged to improve Intelligent Tutoring Systems. We have also added an extra paragraph in the Limitations subsection to describe one of the challenges of building such a system (i.e., high resource consumption) and listed possible solutions.
Minor Recommendations:
- Consider citing the following works to enhance the depth and relevance of your literature review and discussions:
- doi.org/10.3390/bdcc6040130
- doi.org/10.1016/j.eswa.2022.119391
Response: We appreciate the recommendation. We found that the second paper was a good example for our paragraph on hardware resource constraints in the Limitations subsection. We mentioned the work on compressing ResNet-baset networks via layer pruning among other techniques that can be used for reducing the memory footprint of the network.
The quality of the English language used in the manuscript is generally good, with clear sentence structures and appropriate vocabulary. However, there are occasional instances where phrasing can be refined for better clarity and coherence.
Response: Thank you. We have tried to rephrase some of our more convoluted sentences in order to prioritize text clarity.
Round 2
Reviewer 3 Report
The authors have successfully answered to all my questions.